# Efficacy and Safety of Transcranial Direct Current Stimulation on Post-Stroke Dysphagia: A Systematic Review and Meta-Analysis

**DOI:** 10.3390/jcm11092297

**Published:** 2022-04-20

**Authors:** Kelin He, Lei Wu, Yi Huang, Qinqin Chen, Bei Qiu, Kang Liang, Ruijie Ma

**Affiliations:** 1The Third Affiliated Hospital of Zhejiang Chinese Medical University (Zhongshan Hospital of Zhejiang Province), Hangzhou 310005, China; 20175238@zcmu.edu.cn (K.H.); wulei@zcmu.edu.cn (L.W.); liangkang@zcmu.edu.cn (K.L.); 2The Third School of Clinical Medicine (School of Rehabilitation Medicine), Zhejiang Chinese Medical University, Hangzhou 310053, China; 2020111212111224@zcmu.edu.cn (Y.H.); 201911121211153@zcmu.edu.cn (Q.C.); 2020111212111236@zcmu.edu.cn (B.Q.)

**Keywords:** transcranial direct current stimulation, stroke, dysphagia, meta-analysis, systematic review

## Abstract

Dysphagia is one of the most common symptoms in patients after stroke onset, which has multiple unfavorable effects on quality of life and functional recovery. Transcranial direct current stimulation (tDCS) is a non-invasive brain stimulation that is widely used to improve deglutition function. Recently, some studies have confirmed that tDCS enhances deglutition function after stroke. However, the number of evaluation indexes used in those studies was small, and the number of trials included was limited. Most importantly, the optimal stimulation protocol is still uncertain and the safety of tDCS has not been reviewed. Therefore, we conducted a systematic review and meta-analysis to address these shortcomings. Methods: Seven databases were searched entirely, including Pubmed, Cochrane Library, Web of Science, China National Knowledge Infrastructure (CNKI), Chinese Biomedical Literature Service System (SinoMed), Wan-fang database, and the Chinese Scientific Journals Database (VIP) from inception to 31 December 2021. Two reviewers independently evaluated the eligibility of retrieved data according to the selection criteria and assessed the methodological quality of the studies using the Cochrane risk of bias tool. Outcomes, measures, and indicators used in this study included the dysphagia outcome and severity scale (DOSS), modified Mann assessment of swallowing ability (MMASA), functional oral intake scale (FOIS), functional dysphagia scale (FDS), and Kubota’s water-drinking test (KWDT). Sensitivity and subgroup analyses were performed to evaluate the intervention effect more specifically. Results: Fifteen trials with a total of 787 participants (394 subjects in the tDCS groups were treated with true tDCS, and 393 subjects in the control groups were wait-listed or treated with sham tDCS) involving tDCS for dysphagia after stroke and were included in the meta-analysis. Results of this meta-analysis confirmed that tDCS had a positive effect on post-stroke dysphagia. Subgroup analyses revealed that bilateral and high-intensity stimulation with tDCS had a more significant impact on post-stroke dysphagia. Furthermore, no adverse events occurred during the application of tDCS for post-stroke dysphagia. Conclusion: tDCS can promote the recovery of deglutition function in patients with dysphagia after stroke. In addition, bilateral stimulation and high-intensity stimulation may have better effects. However, the safety evidence for tDCS and post-stroke dysphagia is insufficient.

## 1. Introduction

Dysphagia is one of the most common problems following a stroke, with a high incidence of 80% [1]. Although the prevalence of dysphagia gradually decreases over time, 50% of patients still have symptoms of dysphagia at six months after stroke onset [2]. Dysphagia increases the incidence of undernutrition, dehydration, aspiration pneumonia, prolonged hospital stay, and high medical expenses or even death [1,3,4]. Conventional rehabilitation methods include dietary changes, pronunciation training, posture training, oral ice stimulation, swallowing training, and pulse electrotherapy, which are widely used to enhance the recovery of deglutition function. However, conventional rehabilitation may need frequent treatment over many weeks before obtaining a good clinical response, leading to poor patient compliance and, eventually, poor clinical outcomes [5]. In some clinical studies, botulinum toxin injections are also recommended for treating dysphagia [6]. However, this is a treatment option for only few patients suffering from post-stroke dysphagia. Therefore, novel, effective treatment methods are desperately needed to improve swallowing dysfunction after stroke.

To date, there have been several important non-invasive neuromodulations developed to manage neuropsychiatric disease. To be specific, the repetitive transcranial magnetic stimulation (rTMS), theta burst stimulation (TBS, a variant of rTMS), non-invasive vagal nerve stimulation, pharyngeal electrical stimulation (PES), neuromuscular electrical stimulation (NMES), transcranial direct current stimulation (tDCS), and transcranial random noise stimulation (tRNS, a variant of tDCS). Further, there have been several important studies addressing the efficacy and safety of these non-invasive neuromodulations in neuropsychiatric disease, such as dementia and minimal cognitive impairment [7], cognition in brain disorder [8], Alzheimer’s disease [9], and Parkinson’s disease with mild cognitive impairment [10]. Moreover, these methods are also recommended for the treatment of post-stroke dysphagia [11,12,13,14]. Among them, tDCS is a promising adjunct therapy to improve deglutitive function.

As a new non-invasive brain stimulation for managing post-stroke dysphagia, tDCS can trigger and regulate modulating brain activity [15], and has recently drawn widespread attention and is increasingly applied in clinical practice and scientific research. As is well known, tDCS is applied by two surface electrodes (anode and cathode) and works by applying a tiny electrical current (usually 1–2 mA) to a targeted area of the brain. Recently, some reviews have claimed that tDCS improves swallowing dysfunction after stroke [16,17]. However, the number of evaluation indexes used in those reviews was small, and the number of trials included was limited. Most importantly, the safety of the tDCS has not been reviewed for this particular patient cohort [18]. Therefore, we conducted a new systematic review and meta-analysis to further address these problems and shortcomings.

## 2. Materials and Methods

The present study followed the Preferred Reporting Items for Systematic Reviews and Meta-Analyses (PRISMA) guidelines [19], and the study protocol has been registered with PROSPERO (Registration number: CRD42021297331).

### 2.1. Search Strategy

We systematically searched the following seven electronic databases from inception to 31 December 2021: Pubmed, Cochrane Library, Web of Science, CNKI, SinoMed, Wan-fang database, and VIP, to identify all trials of tDCS for treating dysphagia after stroke. The following terms were used as subject words, keywords, free-text terms, and MeSH terms: stroke, cerebral hemorrhage, cerebral infarction, cerebrovascular accident, brain vascular accident, deglutition, dysphagia, swallowing disorder, and transcranial direct current stimulation. Apart from the above, there were no language, region, or countries restrictions.

### 2.2. Eligibility Criteria

This study included all available randomized controlled trials (RCTs) of tDCS for post-stroke dysphagia. Any other types of literature, such as systematic reviews, letters, case reports, editorials, animal studies, commentary, and non-RCTs were excluded.

### 2.3. Participants

A study was included if the adult participants (>18 years of age) were diagnosed with swallowing dysfunction after stroke and the stroke type was either cerebral hemorrhage or cerebral infarction. A study was excluded if dysphagia was caused by traumatic brain injury, oropharyngeal disease, esophageal disease, or mental disorders. In addition, dysphagia associated with neuromuscular disorders was also excluded.

### 2.4. Interventions

The intervention in the experiment group included tDCS alone or in combination with conventional therapy, and the control group included conventional treatment and/or sham tDCS.

### 2.5. Outcomes

The primary outcome indicator for this study was the dysphagia outcome and severity scale, and the secondary outcome indicators included the modified Mann assessment of swallowing ability, functional oral intake scale, functional dysphagia scale, and Kubota’s water-drinking test.

### 2.6. Literature Selection and Data Extraction

One reviewer performed literature searches according to the specified search strategies and downloaded the related citations. All of the selected literature was imported into Endnote X9 (Clarivate, Philadelphia, PA, USA) and duplicate citations were removed using electronic/manual checking. Subsequently, two independent reviewers screened and identified the titles and abstracts of the remaining literature and then independently retrieved the literature that fulfilled the inclusion criteria. Discussion with the corresponding author resolved any inconsistent results between the reviewers. After the initial screenings, two reviewers independently extracted the relevant data from the identified studies. The following information was extracted from each study: general information (authors, publication year), demographic data (sample size, age, gender, stroke onset, stroke type, and stroke location), intervention (tDCS group, control group), tDCS protocol (site of stimulation, intensity of stimulation, duration of stimulation, treatment period), outcome measure (dysphagia outcome and severity scale, modified Mann assessment of swallowing ability, functional oral intake scale, functional dysphagia scale, and Kubota’s water-drinking test), and adverse effects.

### 2.7. Data Analysis

#### 2.7.1. Assessment of Risk of Bias in Included Studies

Two independent reviewers evaluated the risk of bias in each study by using the Cochrane risk of the bias assessment tool [20]. This assessment tool mainly includes seven domains: random sequence generation, allocation concealment, blinding of participants and personnel, blinding of outcome assessment, incomplete outcome data, selective reporting, and other sources of bias. Each domain of the individual study was classified as having a high, low, or unclear risk. Any discordance that occurred between the two reviewers was resolved by discussions with the corresponding author.

#### 2.7.2. Statistical Analysis

All data analyses in this study were conducted with R software (available at: http://www.r-project.org/ (accessed on 10 January 2022), version 3.6.3). Continuous data were calculated as mean differences (MD) with 95% confidence intervals (CI). The I^2^ statistic was used to evaluate the heterogeneity of the studies (with I^2^ statistic > 50% indicating statistically significant heterogeneity). Fixed effects or random-effects models were used according to the heterogeneity (I^2^ statistic > 50%, random effects models; I^2^ statistic < 50%, fixed effects model). In addition, sensitivity analyses and subgroup analyses were carried out to dissect the heterogeneity. A forest plot was used to detect publication bias.

## 3. Results

### 3.1. Literature Selection

A total of 255 published studies were identified (42 references from CNKI, 44 references from Wan-fang, 25 references from VIP, 26 references from SinoMed, 22 references from Pubmed, 44 references from Cochrane Library, 52 references from Web of Science) and imported into Endnote X9. After eliminating duplicates,101 articles remained. We then excluded reviews, case reports, and animal experiments, and 67 studies remained. Mixed interventions, and outcome indicators that did not include dysphagia outcome and severity scale, functional oral intake scale, modified Mann assessment of swallowing ability, functional dysphagia scale, or Kubota’s water-drinking test were also excluded. Finally, 15 trials were considered after reading the full text. A detailed flowchart for the screening process is shown in Figure 1.

### 3.2. Characteristics of Included Studies

A total of 15 articles were included, consisting of 787 patients (393 patients in the control group and 394 patients in the tDCS group) with dysphagia after stroke. The interventions in the control group included CT only (*n* = 7), CT + sham tDCS (*n* = 8), and the intervention in the tDCS group was CT + tDCS. Of these, two studies did not report the course of stroke. The maximum current intensity for tDCS was 2 mA, and the minimum was 1 mA. The shortest treatment period for the intervention was five days, and the longest was two months. For outcome measure, five trials used the dysphagia outcome and severity scale, four trials used the modified Mann assessment of swallowing ability, four trials used the functional oral intake scale, three trials used the functional dysphagia scale, and two trials used the Kubota’s water-drinking test. For adverse effects, six studies reported no adverse events, including skin redness, skin break, epilepsy, seizures, headaches, visual disturbances, skin irritation, or visual disturbance, and the remaining studies provided no information on adverse effects. The detailed characteristics of the included studies are shown in Table 1.

### 3.3. Risk of Bias Assessment

Figure 2 summarizes the risk of bias in the included studies. Six trials reported a method of random sequence generation and were assessed as low risk of bias [21,22,23,24,25,26]; two trials used the wrong randomization method and were evaluated as high risk of bias [27,28]; one trial described the methods of allocation concealment and was regarded as low risk of bias [23]; four trials mentioned the method for blinding the participants and personnel and were considered as low risk of bias [21,29,30,31]; four trials mentioned the method of blinding the outcome assessment and were assessed as low risk of bias [30,31,32,33]; four trials had complete outcome data and were considered as low risk of bias [23,26,29,32]; one trial had incomplete outcome data and was regarded as high risk of bias [31]; one trial did not involve selective reporting and was assessed as low risk of bias [26]. In addition, all trials were not clear about other sources of bias.

### 3.4. Results of the Meta-Analysis

#### 3.4.1. The Meta-Analysis Results for the Dysphagia Outcome and Severity Scale

Five RCTs involved the dysphagia outcome and severity scale [26,27,29,32,33]. After carefully reading the full text of the corresponding studies, the intervention protocols in the included five trials were different. Hence, a subgroup analysis was performed according to the intervention protocol. Since the I^2^ statistic > 50%, a random-effects model was used to perform the meta-analysis. Treatment with tDCS compared with no tDCS showed a significant difference (MD = 1.26, 95% CI = 0.68; 1.84) and the corresponding result is shown in Figure 3A. A subgroup analysis showed that a high stimulation intensity (1.6–2 mA) had a larger positive effect on post-stroke dysphagia than a low stimulation intensity (1–1.5 mA). The corresponding results are shown in Figure 3B. In addition, a sensitivity analysis showed that the results of this meta-analysis were stable.

#### 3.4.2. The Meta-Analysis Results for the Modified Mann Assessment of Swallowing Ability

A total of six studies used the modified Mann assessment of swallowing ability [21,22,24,25,27,28]. Since the I^2^ statistic > 50%, a random-effects model was used to perform the meta-analysis. The results of this meta-analysis showed that when tDCS was compared with no tDCS, there was a significant difference (MD = 7.57, 95% CI = 4.53; 10.62). The corresponding results are shown in Figure 4A. A subgroup analysis showed that bilateral brain stimulation had a larger positive effect (MD = 6.19, 95% CI = 4.65; 7.74) than undamaged brain stimulation (MD = 5.87, 95% CI = 2.40; 9.35). The corresponding results are presented in Figure 4B. In addition, a sensitivity analysis showed that the results of this meta-analysis were stable.

#### 3.4.3. The Meta-Analysis Results for the Functional Oral Intake Scale

Four trials employed the functional oral intake scale [22,23,25,30]. The I^2^ statistic = 31% for this group; thus, a fixed-effects model was used to perform the meta-analysis. This meta-analysis showed a significant difference when tDCS was compared with no tDCS (MD = 0.64, 95% CI = 0.52; 0.77) and the corresponding results are presented in Figure 5A. A subgroup analysis result showed that stimulation of the bilateral brain had a positive effect (MD = 0.86, 95% CI = 0.26; 1.46). Stimulation of the undamaged brain had a moderate positive effect (MD = 0.48, 95% CI = 0.21; 0.75). The corresponding results are shown in Figure 5B. In addition, sensitivity analysis showed that the results of this meta-analysis were credible.

#### 3.4.4. The Meta-Analysis Results of the Functional Dysphagia Scale

Three studies used the functional dysphagia scale [23,26,31]. Since the I^2^ statistic = 0%, a fixed model was used to perform the meta-analysis. The results revealed that when tDCS was compared with no tDCS, there was a significant difference (MD = −8.15, 95% CI = -13.03; –3.27) and the corresponding results are shown in Figure 6. In addition, sensitivity analysis showed that the trial by Wang (2020) was a major source of heterogeneity. After removing this study, the MD for the functional dysphagia scale was −6.30 [95% CI: −12.74; 0.14, *p* = 0.0553].

#### 3.4.5. The Meta-Analysis Results for Kubota’s Water-Drinking Test

Two trials involved Kubota’s water-drinking test [24,34]. Since the I^2^ statistic = 95%, a random-effects model was used to perform a meta-analysis. When tDCS was compared with no tDCS, there was a significant difference (MD = 0.93, 95% CI = 0.25; 1.61) and the corresponding results are shown in Figure 7. In addition, sensitivity analysis showed that the results of this meta-analysis were credible.

#### 3.4.6. The Safety of tDCS

Six studies specified that no skin redness, skin breaks, epilepsy, seizures, headaches, visual disturbances, skin irritation, visual disturbances, or serious adverse events (severe or medically significant but not immediately life-threatening events, include the requirement for inpatient hospitalization or prolongation of hospitalization) occurred [22,26,29,30,31,32]. The one trial that used the highest intensity stimulation (2 mA) claimed no adverse effects occurred [29]. The other trials did not mention any adverse events.

#### 3.4.7. Publication Bias

Publication bias is a potential concern when interpreting the meta-analysis results. In this study, funnel plots were used to assess publication bias. A publication bias is indicated by an asymmetrical funnel around the pooled effect size. The selected studies did not lie symmetrically around the pooled effect size, as shown in Figure 8, Figure 9, Figure 10, Figure 11 and Figure 12.

## 4. Discussion

Overall, our analysis based on primary outcome measures demonstrated that anodal tDCS has a beneficial effect on post-stroke dysphagia, and this result was consistent with previously published studies [16,17]. Moreover, a high intensity, bilateral stimulation tDCS protocol may have a better effect. In contrast to previous studies, our study contains all published trials up to 31 December 2021, except for those for which we could not obtain critical and essential outcomes, or the use of electroacupuncture in the control group did not meet our inclusion criteria [35,36]. Many kinds of swallowing function rating scales are used in clinics, such as the dysphagia outcome and severity scale, modified Mann assessment of swallowing ability, functional oral intake scale, functional dysphagia scale, and Kubota’s water-drinking test. In this study, we used the dysphagia outcome and severity scale as the primary outcome indicator because it has high reliability [37]. To some extent, our study is valuable because we have included new studies, and our results have updated the stimulation protocol for tDCS. Our most critical finding is that we have used different swallow-related scales to evaluate the effect size of tDCS. In addition, we also reviewed the adverse effects of tDCS on post-stroke dysphagia.

It is currently believed that different polarity, current, or target brain regions for tDCS management would contribute to a wide variety of effects. It has been proposed that tDCS induces neuroplastic changes in motor cortical excitability, i.e., anodal tDCS induces sustained elevations in neural cell membrane potentials, and cathodal tDCS induces sustained decreases in neural cell membrane potentials [38]. In addition, the effect of tDCS may vary according to target brain regions, i.e., the same anodal stimulation may depolarize or hyperpolarize depending on whether the target is in the gyri or sulci, which may explain the large inter-individual variability in tDCS responses [39]. However, recently a new hypothesis has addressed the different effects of polarity, current, and target brain regions on tDCS management that is referred to as the neural noise hypothesis [40,41]. To be specific, the after-effect of tDCS might depend on the overall glutamatergic, GABAergic, dopaminergic, and serotoninergic synaptic activity. Therefore, analysis of the intervention plan of tDCS on post-stroke dysphagia is quite necessary. Thus, we analyzed the effect of stimulation site, stimulus intensity, and other aspects of tDCS on post-stroke dysphagia. The specific details are listed below.

### 4.1. Effect of Stimulation Site of tDCS on Post-Stroke Dysphagia

Subgroup analyses of the modified Mann assessment of swallowing ability and the functional oral intake scale demonstrated that anodal tDCS of the damaged hemisphere and bilateral hemispheres could significantly affect deglutition function in stroke patients. When tDCS is used in different brain areas, it can give rise to various manifestations, for instance, changes in brain networks, cognitive performance, and brain metabolite and neurotransmitter levels [42,43,44]. Since swallowing has bilateral hemispheric representation, the reorganization of the damaged cerebral hemisphere may also play an essential role in recovering deglutition function after stroke [45,46]. For the meta-analysis results of the modified Mann assessment of swallowing ability, the weighted effect size for the bilateral hemisphere was large at 6.19 compared to a medium effect size of 5.87 for the undamaged hemisphere. For the meta-analysis results of the functional oral intake scale, the weighted effect size for the bilateral hemisphere was large at 0.86 compared to the medium effect size of 0.48 for the undamaged hemisphere. These results suggests that anodal tDCS of the bilateral hemisphere is superior to the undamaged hemispheres for improving deglutition function after stroke. Our results are consistent with previous studies showing that the application of tDCS to the bilateral hemisphere may have some inherent advantages over applying it to the undamaged hemisphere. Bilateral tDCS can affect neuronal activity and connectivity within and across the sensorimotor cortical network in the brain [47]. Of course, this result requires further RCT confirmation.

### 4.2. Effect of Intense Stimulation of tDCS on Post-Stroke Dysphagia

The subgroup analysis of the dysphagia outcome and severity scale demonstrated that both low and high-intensity stimulation with anodal tDCS can significantly affect deglutition function in stroke patients. Notably, high-intensity stimulation with anodal tDCS has more advantages than low-intensity stimulation for improving deglutition function after stroke. It is generally known that tDCS is a non-invasive technique that uses a constant, low-intensity direct current (1~2 mA) to regulate neuronal activity in the cerebral cortex. Previous studies had identified that high-intensity (2 mA) stimulation with tDCS had a greater effect on neural plasticity than low-intensity (1 mA) stimulation [48,49]. Here, we divided stimulation with tDCS into high intensity (1.6–2 mA) or low intensity (1–1.5 mA) according to the included studies’ characteristics. The results showed that high-intensity stimulation has a better effect size than low-intensity stimulation. As described in previous studies [50], high-intensity stimulation resulted in a significant increase of motor-evoked potentials amplitudes, whereas low-intensity stimulation is also associated with less variability in corticospinal excitability. Moreover, higher cortical excitability is associated with better swallowing function recovery [51].

### 4.3. Duration Stimulation of tDCS

Apart from the intensity of stimulation by tDCS, the duration of stimulation is also an element that can have an impact on the efficacy of tDCS [52]. It has been shown in human studies that tDCS duration varied from 3 to 40 min [53]. In our research, we found that the longest duration of stimulation was 40 min. However, too few durations exist for tDCS for post-stroke dysphagia and it was hard to apply subgroup analyses. Therefore, we were not able to investigate the optimal duration of stimulation.

### 4.4. Treatment Period of tDCS

In this study, the treatment period for tDCS differed between studies. In a previous study, multiple stimulation with tDCS per week may produce a cumulative effect on brain activity and increase its impact on behavioral outcomes [54]. It is generally thought that the short-term and long-term effects of tDCS are different, one of which is resting membrane potential depolarization through non-synaptic mechanisms [55], and the other is N-methyl-D-aspartate-dependent mechanisms [56].

### 4.5. Adverse Effects of tDCS on Post-Stroke Dysphagia

We should recognize that for any stimulation protocol there exists a certain degree of risk that could cause problems in particular cases. Many questions remain open until extensive research or clinical experience is gained. In general, low intensity (1–2 mA) tDCS is considered safe [18]. However, this evidence was collected mainly from healthy subjects and neurological and psychiatric patients. In our research, some studies expressly affirmed that there were no adverse effects reported for post-stroke dysphagia. It should be noted explicitly that one study that used the highest intensity (2 mA) for tDCS declared no adverse events occurred [29]. However, a large sample study of tDCS of healthy subjects and other diseases has reported some negative effects, such as pain, fatigue, itching, etc. [57]. Thus, small sample sizes may explain why the studies included in this review did not report adverse effects.

### 4.6. Limitation

Firstly, this study’s data on adverse events was relatively small. As a result, we could not create a quantitative analysis based on the available data; therefore, we could only conduct a narrative review for this study. Secondly, some aspects of the stimulation protocol were different, such as the duration of stimulation and the course of stimulation. Therefore, meta-regression may be needed to adjust these variables. However, the small sample size for those studies limited our ability to do so.

## 5. Conclusions

The application of tDCS can promote the recovery of deglutition function in patients with dysphagia after stroke, and bilateral stimulation and high-intensity stimulation may have better effects. However, the safety evidence of tDCS for post-stroke dysphagia is insufficient. In addition, all studies are single-center and lack a unified evaluation scale. Therefore, future research should take steps in this direction to solve these deficiencies.

## Figures and Tables

**Figure 1 jcm-11-02297-f001:**
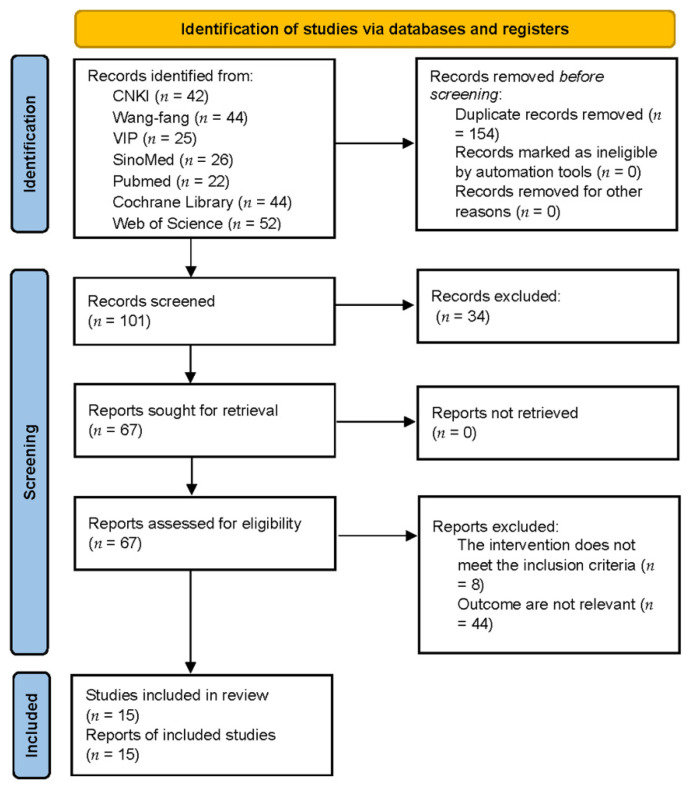
Detailed flowchart of the screening process. CNKI, China National Knowledge Infrastructure; VIP, the Chinese Scientific Journals Database; SinoMed, Chinese Biomedical Literature Service System; *n*, number of publications.

**Figure 2 jcm-11-02297-f002:**
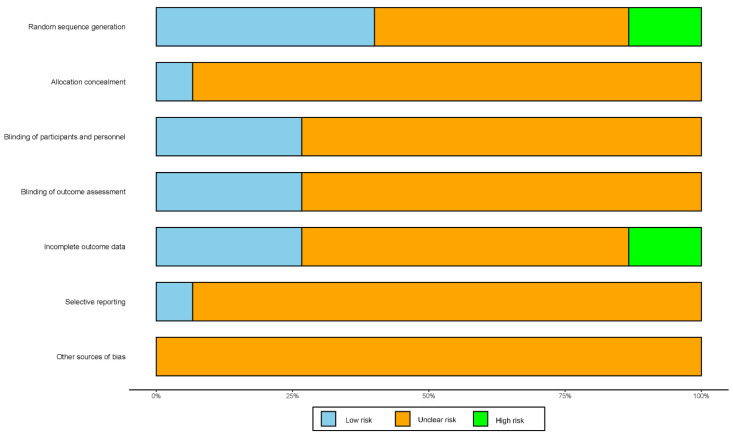
Quality evaluation of the risk of bias in the included literature.

**Figure 3 jcm-11-02297-f003:**
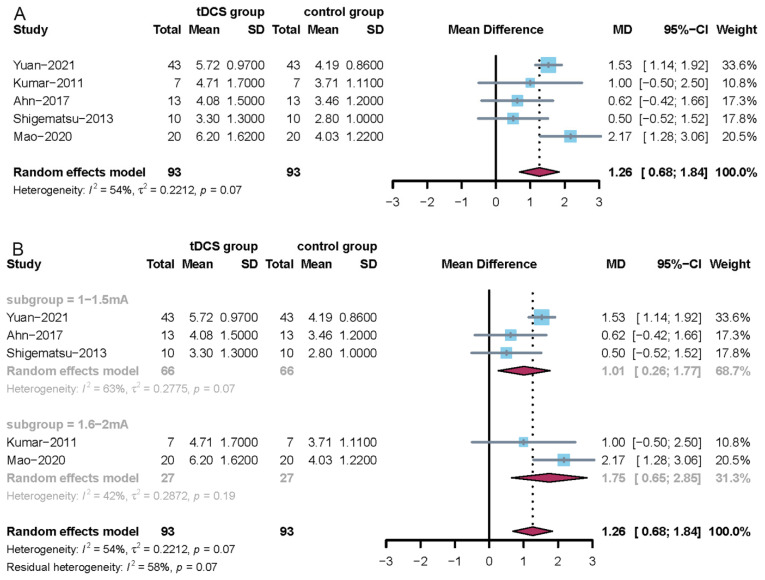
Forest plots of the dysphagia outcome and severity scale. (**A**) Overall analysis of the five included trials; (**B**) subgroup analysis based on the stimulation intensity (1–1.5 mA vs. 1.6–2 mA).

**Figure 4 jcm-11-02297-f004:**
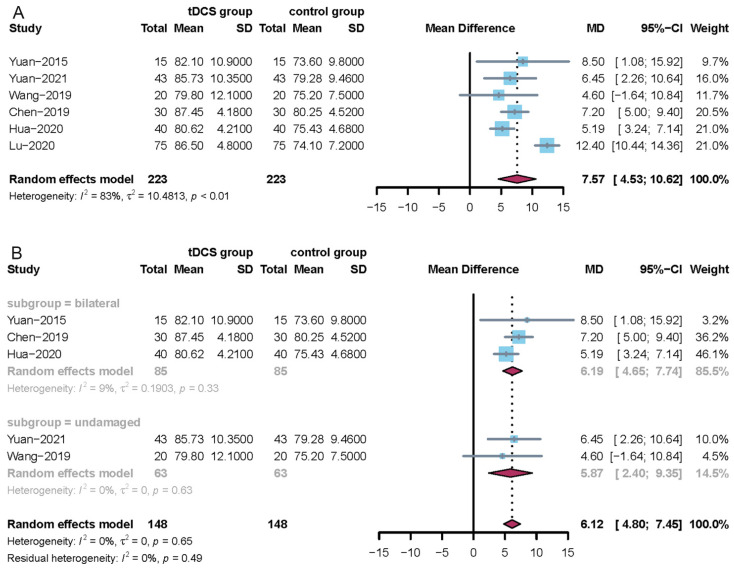
Forest plots of the modified Mann assessment of swallowing ability. (**A**) Overall analysis of the six included trials; (**B**) subgroup analysis using tDCS on the bilateral or undamaged brain.

**Figure 5 jcm-11-02297-f005:**
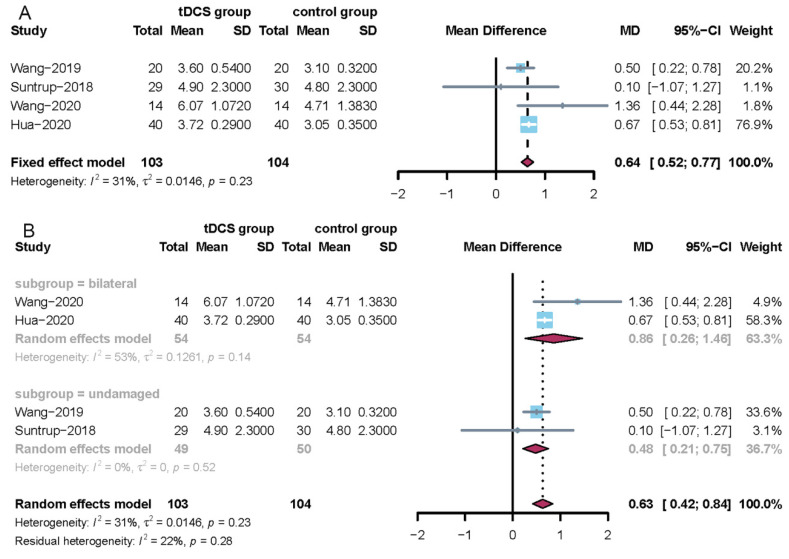
Forest plots for the functional oral intake scale. (**A**) Overall analysis of four included trials; (**B**) subgroup analysis using tDCS on the bilateral or undamaged brain.

**Figure 6 jcm-11-02297-f006:**
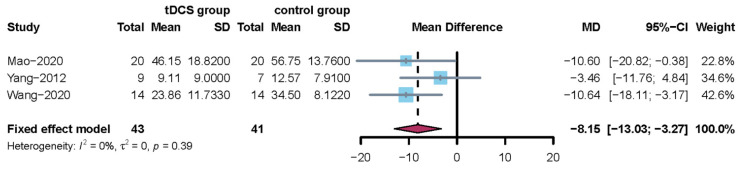
Forest plot for the functional dysphagia scale (overall analysis of three included trials).

**Figure 7 jcm-11-02297-f007:**
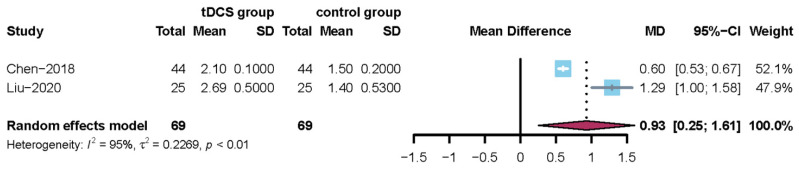
Forest plot for the Kubota’s water-drinking test (overall analysis of two included trials).

**Figure 8 jcm-11-02297-f008:**
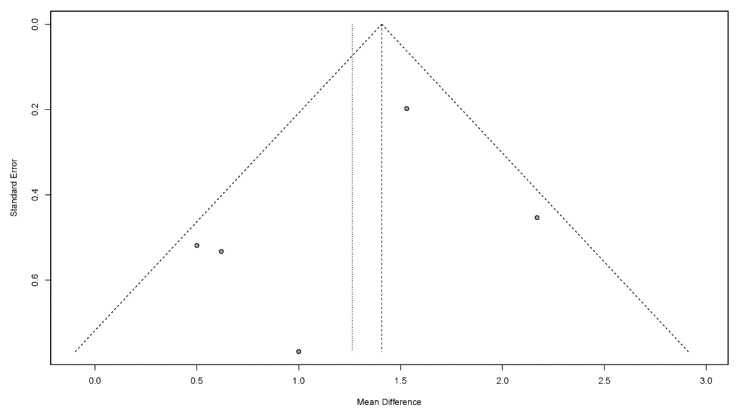
Funnel plots for the dysphagia outcome and severity scale.

**Figure 9 jcm-11-02297-f009:**
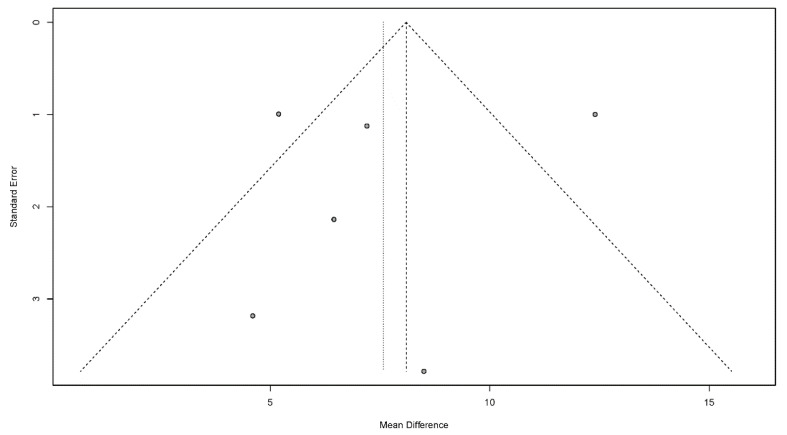
Funnel plots for the modified Mann assessment of swallowing ability.

**Figure 10 jcm-11-02297-f010:**
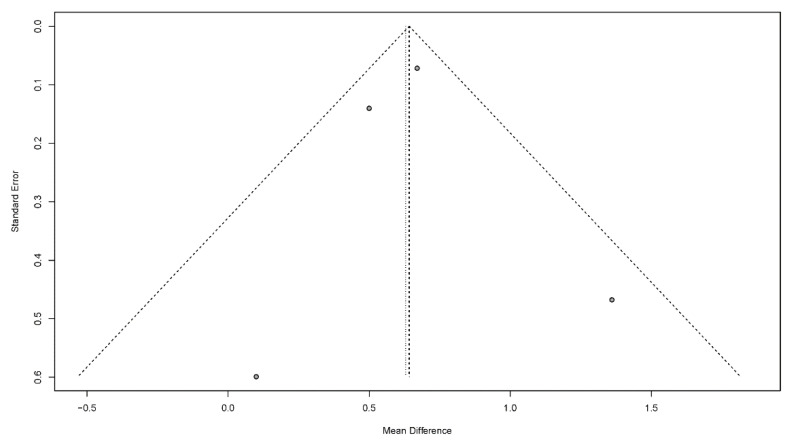
Funnel plots for the functional oral intake scale.

**Figure 11 jcm-11-02297-f011:**
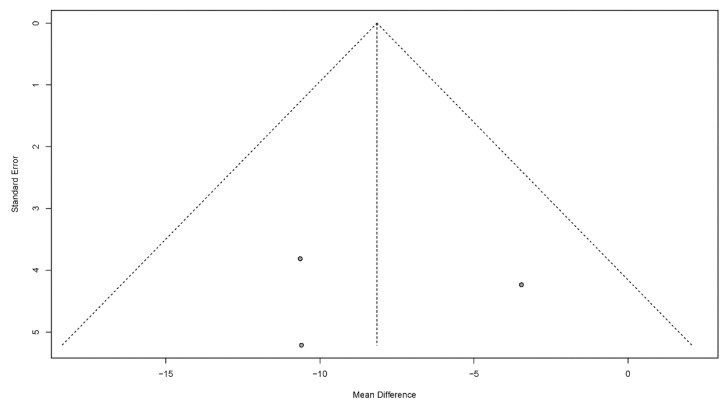
Funnel plots for the functional dysphagia scale.

**Figure 12 jcm-11-02297-f012:**
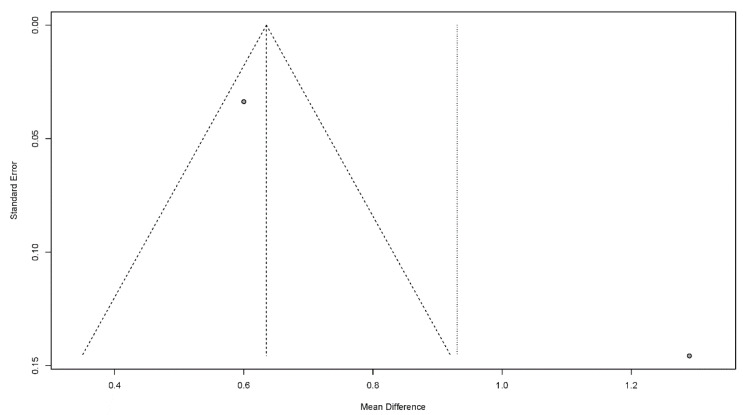
Funnel plots for the Kubota’s water-drinking test.

**Table 1 jcm-11-02297-t001:** Characteristics of included studies.

Study	Publish Year	Sample		Age (Years)		Gender (M/F)		Stroke Onset		Stroke Type	Stroke Location	Intervention		tDCS Protocol				Outcome Measure	Adverse Effect
		Control Group	tDCS Group	Control Group	tDCS Group	Control Group	tDCS Group	Control Group	tDCS Group			Control Group	tDCS Group	Site of Stimulation	Intensity of Stimulation	Duration of Stimulation	Treatment Period		
Yuan	2015	15	15	57.4 ± 7.2	60.7 ± 11.5	13/2	14/1	(58.5 ± 28.5) d	(57.7 ± 25.8) d	CH + CI	Cerebellum	CT + sham tDCS	CT + tDCS	Bilateral cerebellar hemisphere	1 mA	20 min	Once a day for 20 days	MMASA	Unclear
Yuan	2021	43	43	62.95 ± 5.74	61.72 ± 5.29	27/16	25/18	(1.79 ± 0.57) m	(1.64 ± 0.49) m	CI	Unclear	CT	CT + tDCS	Undamaged pharyngeal motor cortex	1.5 mA	20 min	Five times per week for four weeks	MMASA, DOSS	Unclear
Wang	2019	20	20	60.8 ± 11.2	64.8 ± 7.2	15/5	13/7	(47.9 ± 21.6) d	(51.2 + 28.9) d	CH + CI	Basal ganglia	CT + sham tDCS	CT + tDCS	Undamaged pharyngeal motor cortex	1.5 mA	20 min	Five times per week for two weeks	MMASA, FOIS	No
Kumar	2011	7	7	70 ± 11.96	79.71 ± 10.21	4/3	3/4	(96.71 ± 45.93) h	(80.29 ± 42.31) h	CI	Unilateral hemisphere	CT + sham tDCS	CT + tDCS	Undamaged hemisphere	2 mA	30 min	Once a day for five days	DOSS	No
Ahn	2017	13	13	66.38 ± 10.67	61.62 ±10.28	6/7	9/4	(11.62 ± 4.56) m	(12.27 ± 4.92) m	CI	Unilateral cortical or subcortical	CT + sham tDCS	CT + tDCS	Bilaterally pharyngeal motor cortex	1 mA	20 min	Five times per week for two weeks	DOSS	No
Mao	2020	20	20	61.25 ± 8.02	59.80 ± 7.27	8/12	11/9	(3.60 ± 2.49) m	(3.25 ± 2.24) m	CH + CI	Brain stem	CT	CT + tDCS	Undamaged pharyngeal motor cortex	1.6 mA	20 min	Six times per week for eight weeks	DOSS, FDS	No
Shigematsu	2013	10	10	64.7 ± 8.9	66.9 ± 6.3	7/3	7/3	at least 1 month		CH + CI	Unclear	CT + sham tDCS	CT + tDCS	Affected pharyngeal motor cortex	1 mA	20 min	10 days	DOSS	Unclear
Suntrup	2018	30	29	67.2 ± 14.5	68.9 ± 11.5	17/13	17/12	(116.8 ± 64.9) h	(116.3 ± 98.9) h	CI	Supratentorial; infratentorial	CT + sham tDCS	CT + tDCS	Unaffected pharyngeal motor cortex	1 mA	20 min	Once a day for four days	FOIS, FEDSS	No
Wang	2020	14	14	62.00 ± 10.46	61.43 ± 11.24	10/4	11/3	(67.50 ± 47.62) d	(66.79 ± 38.62) d	CH + CI	Brainstem	CT + sham tDCS	CT + tDCS	Bilateral oesophageal coritical area	1 mA	40 min	Five times per week for four weeks	FOIS, FDS	Unclear
Yang	2012	7	9	70.57 ± 8.46	70.44 ± 12.59	3/4	6/3	(26.9 ± 7.8) d	(25.2 ± 11.5) d	CI	Unilateral hemisphere	CT + sham tDCS	CT + tDCS	Affected pharyngeal motor cortex	1 mA	20 min	Five times per week for two weeks	FDS	No
Chen	2018	44	44	67.8 ± 1.8	68.6 ± 1.5	24/20	23/21	not mentioned		CH + CI	Unclear	CT	CT + tDCS	Damaged hemisphere of the oropharyngeal cortex	1.2 mA	20 min	Five times per week for two weeks	KWDT	Unclear
Chen	2019	30	30	62.93 ± 4.12	61.27 ± 4.52	19/11	17/13	(1.92 ± 0.24) m	(1.89 ± 0.17) m	CH + CI	Unclear	CT	CT + tDCS	Bilateral pharyngeal sensory-motor cortex	1.4 mA	20 min	Five times per week for two weeks	MMASA	Unclear
Hua	2020	40	40	61.28 ± 10.15	60.29 ± 9.48	29/11	31/9	(48.16 ± 9.97) d	(47.39 ± 10.83) d	CH + CI	Basal ganglia	CT	CT + tDCS	Bilateral pharyngeal sensory-motor cortex	1 mA	20 min	Twice a day, ten times per week for four weeks	MMASA, FIOS	Unclear
Liu	2020	25	25	54.92 ± 3.82	55.82 ± 3.74	15/10	14/11	(14~90) d		CH + CI	Unclear	CT	CT + tDCS	Damaged pharyngeal cortex	1.2 mA	20 min	Five times per week for two weeks	KWDT	Unclear
Lu	2020	75	75	57.3 ± 2.2	57.5 ± 2.1	36/39	35/40	(48.3 ± 2.5) d	(48.5 ± 2.4) d	CI	Unclear	CT	CT + tDCS	Damaged oropharyngeal cortex	1.2 mA	20 min	Five times per week for two weeks	MMASA	Unclear

Abbreviation: d, day; h, hour; m, month; CI, cerebral infarction; CH, cerebral hemorrhage; CT, conventional treatment; DOSS, dysphagia outcome and severity scale; MMASA, modified Mann assessment of swallowing ability; FOIS, functional oral intake scale; FDS, functional dysphagia scale; KWDT, Kubota’s water-drinking test; FEDSS, fiberoptic endoscopic dysphagia severity scale. Adverse effects: skin redness, skin break, epilepsy, seizures, headaches, visual disturbances, skin irritation, or visual disturbance.

## Data Availability

All data relevant to the study are included in the article or uploaded as Appendix A.

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
