# Peer review of "Efficacy and Safety of Transcranial Direct Current Stimulation on Post-Stroke Dysphagia: A Systematic Review and Meta-Analysis"

_jcm, 2022, doi:10.3390/jcm11092297_

Round 1
Reviewer 1 Report
This is a systematic review and meta-analysis on the effects of tDCS on swallowing function in patients suffering from post-stroke dysphagie. The topic is of interest due to its clinical relevance for a large number of patients suffering from this condition, particularly during the acute phase of stroke but similarly during the chronic stage and therapeutic options are limited. tDCS is among other neuromodulatory treatment approaches a very promising, non-invasive treatment option that can be easily be applied in patient to improve swallowing function and thereby reducing the risk of dysphagia-related sequelae, such as aspiration pneumonia, malnutrition and an overall decreased quality of life. Optimal stimulation targets (contralesional vs. Ipsilesional vs. bihemispheric) and optimal stimulation intensity are not clear yet, making such a review welcome for reconsideration.
Structure and language of this manuscript are mainly sound, relevant studies were included in this review. However, there are a number of concerns that should be addressed:
Main concerns:
- Introduction: Botulinumtoxin is addressed, however this is a treatment option for only few patients suffering from post-stroke dysphagia. This should be specified. Furthermore, other neuromodulatory treatment options, particularly Pharyngeal Electrical Stimulation, but also NMES and rTMS should at least be mentioned shortly described (e.g. findings from the PHAST-TRAC trial)
- Introduction: It is stated that safety of tDCS has not been reviewed. This should be specified – it has not been reviewed for this particular patient cohort. In general, low intensity tDCS is considered to be safe, e.g. Antal, A. et al. Low intensity transcranial electric stimulation: Safety, ethical, legal regulatory and application guidelines. Clin Neurophysiol. 2017 Sep; 128(9). I think, since this is a relevant aspect of this manuscript, this should also be discussed more specifically in the discussion section of the article.
- Table 1: please consider rephrasing: undamaged pharyngeal motor cortex regarding stimulation site in patients who had brainstem infarction – this is not specific.
- Table 1: adverse effects: unclear – please define.
- A major constraint in most of these studies is a lack of instrumental evaluation of swallowing function – this should be discussed. Furthermore, Suntrup-Krueger et al. Used an instrument-based score called FEDSS as well as the FOIS which may be implemented in table 1 as well.
- Safety: Please definde what would be considered other SAEs.
- I believe that the discussion would benefit from a more detailed discussion of the physiological rationale behind the different stimulation modes, e.g. not only providing a statement (e.g. „The results showed that high-intensity stimulation has a better effect size than low-intensity stimulation.“) but also giving a possible hypothesis for this. This may also imply comparing explanations for certain stimulation parameters chosen by the respective study group.
Minor concerns:
- ll 42: dysphagia incidence during the acute stage of stroke is considered to be even higher (up to 81%, see: Arnold M, Liesirova K, Broeg-Morvay A, et al. Dysphagia in acute stroke: incidence, burden and impact on clinical outcome. PLoS One 2016;11:e0148424.)
- Data analysis: It is stated that any discordance that occurred between the two reviewers was resolved by discussions with the corresponding author. How often did this take place?
- Figure 1 should be reformatted (e.g. there should not be a space after each number before the bracket à instead of: (n=22 ) it should be (n = 22) or (n=22). Furthermore, the number of cases should
- Figures 3 – 7: These tables are a little unclear and not easily accessable. Please give each subgroup-analysis a clear heading and provide precise descriptions in the table legend.
Line 128: „In our study“ may be replace by: Here,(…)
Author Response
Dear Reviewer,
We are truly grateful for the valuable comments and suggestions from the reviewers. Here we submit a new version of our manuscript with the title " Efficacy and Safety of Transcranial Direct Current Stimulation on Post-stroke Dysphagia:A Systematic Review and Meta-Analysis”, which has been carefully modified accordingly. According to your guidance, we have revised our manuscript again very carefully. All changes have been highlighted in red color in the revised manuscript. In addition, we provided an itemized, point-by-point response to your comments in the second round.
Main concerns:
- Introduction: Botulinumtoxin is addressed, however this is a treatment option for only few patients suffering from post-stroke dysphagia. This should be specified. Furthermore, other neuromodulatory treatment options, particularly Pharyngeal Electrical Stimulation, but also NMES and rTMS should at least be mentioned shortly described (e.g. findings from the PHAST-TRAC trial)
Response: Thank you for your professional review of our article. I totally agree. (Page 2, Line: 54-55, Line:58-60)
- Introduction: It is stated that safety of tDCS has not been reviewed. This should be specified – it has not been reviewed for this particular patient cohort. In general, low intensity tDCS is considered to be safe, e.g. Antal, A. et al. Low intensity transcranial electric stimulation: Safety, ethical, legal regulatory and application guidelines. Clin Neurophysiol. 2017 Sep; 128(9). I think, since this is a relevant aspect of this manuscript, this should also be discussed more specifically in the discussion section of the article.
Response: thank you for your careful review, we had changed the expression as follow: the safety of the tDCS has not been reviewed for this particular patient cohort. In addition, we had discussed in the discussion section of the article. (Introduction section, Page 2, Line: 76-77; Discussion section, Page 8, Line:185-189).
- Table 1: please consider rephrasing: undamaged pharyngeal motor cortex regarding stimulation site in patients who had brainstem infarction – this is not specific.
Response: We thank the reviewer for the professional comments and valuble suggestions. I totally agree.
- Table 1: adverse effects: unclear – please define.
Response: Thanks the review for the valuable comments. Adverse effects included skin redness, skin break, epilepsy, seizures, headaches, visual disturbances, skin irritation, or visual disturbance. (Table 1, Page 8)
- A major constraint in most of these studies is a lack of instrumental evaluation of swallowing function – this should be discussed. Furthermore, Suntrup-Krueger et al. Used an instrument-based score called FEDSS as well as the FOIS which may be implemented in table 1 as well.
Response: thank you for your careful review. we had added the corresponding content in table 1. (Table 1, Page 8)
- Safety: Please definde what would be considered other SAEs.
Response: Thank you for your careful review. According to previous study, serious AEs are severe or medically significant but not immediately life-threatening events, include the requirement for inpatient hospitalization or prolongation of hospitalization. ( 3.4.6 The safety of tDCS, Page 5, Line:85-86)
- I believe that the discussion would benefit from a more detailed discussion of the physiological rationale behind the different stimulation modes, e.g. not only providing a statement (e.g. „The results showed that high-intensity stimulation has a better effect size than low-intensity stimulation.“) but also giving a possible hypothesis for this. This may also imply comparing explanations for certain stimulation parameters chosen by the respective study group.
Response: This is a very important suggestion. We had added a possible hypothesis for different stimulation. (Discussion section, Page 8, Line:166-169)
Minor concerns:
- ll 42: dysphagia incidence during the acute stage of stroke is considered to be even higher (up to 81%, see: Arnold M, Liesirova K, Broeg-Morvay A, et al. Dysphagia in acute stroke: incidence, burden and impact on clinical outcome. PLoS One2016;11:e0148424.)
Response: Thank you for your helpful guidance. We have updated the incidence of post-stroke dysphagia. (Introduction section, Page:2, Line:44)
- Data analysis: It is stated that any discordance that occurred between the two reviewers was resolved by discussions with the corresponding author. How often did this take place?
Response: We have trained the two reviewers in the early stage. Then, in the formal implementation stage, for example, if 100 articles are included, there are 8 articles with inconsistent scores. This part of the inconsistency is discussed with the corresponding author.
- Figure 1 should be reformatted (e.g. there should not be a space after each number before the bracket à instead of: (n=22 ) it should be (n = 22) or (n=22). Furthermore, the number of cases should
Response: Thank you for your guidance in manuscript preparation. We have made the corresponding changes.
- Figures 3 – 7: These tables are a little unclear and not easily accessable. Please give each subgroup-analysis a clear heading and provide precise descriptions in the table legend.
Response: This is a very important suggestion. We have made the corresponding changes. (Result section, Page:2-5)
- Line 128: „In our study“ may be replace by: Here,(…)
Response: Thank you for your helpful guidance. We have made the corresponding correction. (Discussion section, Page:8, Line:163)

Reviewer 2 Report
Manuscript is properly done with a lot of important points that were nicely addressed. I think if we can find more evidence related to tDCS for post stroke dysphagia it it will be really helpful for the patientsAuthor Response
Thank you for your careful review.
Reviewer 3 Report
General comments:
This is an important and timely meta-analysis. I appreciate the issue the authors raised. I only had some minimal comments to be addressed.
1. The authors said “…and 393 subjects in control groups…” in the abstract. I am not sure what the “control” is. Is it sham-control or waiting list? Please clarify it.
2. The same in the section of abstract, the authors said “… no adverse occurred during the application…”. It might be “… no adverse events occurred during the application…”.
3. The authors had provided some important introduction about the post-stroke dysphagia. I appreciated with their efforts regarding the introduction of post-stroke dysphagia. However, the introduction about the non-invasive neuromodulation and tDCS were too few to emphasize these novel therapeutic interventions. To date, there have been several important non-invasive neuromodulation developed to manage neuropsychiatric disease. To be specific, the repetitive transcranial magnetic stimulation (rTMS), theta burst stimulation (TBS, one of the variant of rTMS), non-invasive vagal nerve stimulation, tDCS, and transcranial random noise stimulation (tRNS, one of the variant of tDCS). Further, there have been several important meta-analyses/network meta-analyses/randomized controlled trials (RCTs) addressing the efficacy and safety of those non-invasive neuromodulation in neuropsychiatric disease, such as dementia/minimal cognitive impairment (PMID: 30229671), cognition in brain disorder (PMID: 33070785), Alzheimer's disease (PMID: 33115936), and minimal cognitive impairment + Parkinson (PMID: 33408684). Therefore, I would strongly recommend the authors to cite all these references and make a brief statement about the WIDE application of non-invasive neuromodulation in neuropsychiatric disease. That will bring the authors a more comprehensive and wide knowledge about the non-invasive neuromodulation available.
4. The authors had followed the PRISMA guideline. It is good. However, since the latest PRISMA version (PRISMA 2020) had been published, I would recommend the authors to use the latest PRISMA 2020 (PMID: 33782057).
5. The authors said “Literature was included in which adult participants (>18 years of age) were diagnosed with swallowing dysfunction after stroke and in whom the stroke type was either cerebral hemorrhage or cerebral infarction.” Did the authors restricted to stroke at any specific brain regions?
6. Since there were several different rating scales to evaluate the severity of dysphagia and other target outcomes (their primary and secondary outcome), the choice of mean difference might not be suitable. To be specific, the mean difference was only suitable for situation that all the included studies applied the same rating scales (or the same units). In contrary, in situation of different rating scales, the standardized mean difference would be a good choice, which could be done by R.
7. I noticed that all the included RCTs applied different polarity of tDCS with different currents. The different polarity and different currents of tDCS might exert different effect (i.e. enhancing or inhibiting effect) to the targeted brain. Please address this issue and how the authors resolve this heterogeneity.
8. I did not see the figure legend of Figures 3, 4, 5, 6, and 7. Please add it.
9. Where is the evaluation of publication bias (i.e. funnel plot or Egger’s test)?
10. Finally, echoing the comment 7, the different polarity/current/target brain regions would contribute to wide variety of effect. However, recently there is another new-raised hypothesis to address the different polarity/current/target brain regions of tDCS management. That is neural noise hypothesis. Please check the reference PMID: 21685932, and PMID: 33691622 and make a brief discussion about this.
11. Finally, per the PRISMA guideline (either 2009 or 2020 version), there are several important items missing. For example, the PRISMA checklist, statement of original authors contact, keywords used in each database, and reasons for exclusion of each study (mandatory for 2020 version). Please revise accordingly.
Author Response
We are truly grateful for the valuable comments and suggestions from the reviewers. Here we submit a new version of our manuscript with the title " Efficacy and Safety of Transcranial Direct Current Stimulation on Post-stroke Dysphagia:A Systematic Review and Meta-Analysis”, which has been carefully modified accordingly. According to your guidance, we have revised our manuscript again very carefully. All changes have been highlighted in red color in the revised manuscript. In addition, we provided an itemized, point-by-point response to your comments in the second round.
General comments:
This is an important and timely meta-analysis. I appreciate the issue the authors raised. I only had some minimal comments to be addressed.
- The authors said “…and 393 subjects in control groups…” in the abstract. I am not sure what the “control” is. Is it sham-control or waiting list? Please clarify it.
Response: Thank you for your careful review. Patients in the tDCS groups were treated with true tDCS, while the control group were treated with wait list or sham tDCS. We have made corrections accordingly. (Page:1, Line:30-31)
2. The same in the section of abstract, the authors said “… no adverse occurred during the application…”. It might be “… no adverse events occurred during the application…”.
Response: Thank you for your helpful review. We have made corrections accordingly. (Page:1, Line:35)
3. The authors had provided some important introduction about the post-stroke dysphagia. I appreciated with their efforts regarding the introduction of post-stroke dysphagia. However, the introduction about the non-invasive neuromodulation and tDCS were too few to emphasize these novel therapeutic interventions. To date, there have been several important non-invasive neuromodulation developed to manage neuropsychiatric disease. To be specific, the repetitive transcranial magnetic stimulation (rTMS), theta burst stimulation (TBS, one of the variant of rTMS), non-invasive vagal nerve stimulation, tDCS, and transcranial random noise stimulation (tRNS, one of the variant of tDCS). Further, there have been several important meta-analyses/network meta-analyses/randomized controlled trials (RCTs) addressing the efficacy and safety of those non-invasive neuromodulation in neuropsychiatric disease, such as dementia/minimal cognitive impairment (PMID: 30229671), cognition in brain disorder (PMID: 33070785), Alzheimer's disease (PMID: 33115936), and minimal cognitive impairment + Parkinson (PMID: 33408684). Therefore, I would strongly recommend the authors to cite all these references and make a brief statement about the WIDE application of non-invasive neuromodulation in neuropsychiatric disease. That will bring the authors a more comprehensive and wide knowledge about the non-invasive neuromodulation available.
Response: Thank you for providing us with a clear guidance about what we need to do to improve our manuscript.We have modified this section in full agreement to the reviewer’s indications.(Introduction section, Page:2, Line:57-68)
- The authors had followed the PRISMA guideline. It is good. However, since the latest PRISMA version (PRISMA 2020) had been published, I would recommend the authors to use the latest PRISMA 2020 (PMID: 33782057).
Response: Thank you for your helpful review. We agree with this suggestion, and cited the latest PRISMA guideline (2020).
5. The authors said “Literature was included in which adult participants (>18 years of age) were diagnosed with swallowing dysfunction after stroke and in whom the stroke type was either cerebral hemorrhage or cerebral infarction.” Did the authors restricted to stroke at any specific brain regions?
Response: We did not restrict to stroke at any specific brain regions. Disorders of swallowing in stroke (Ischemic or hemorrhagic stroke by computed tomography (CT) or magnetic resonance imaging (MRI)) were included.
6. Since there were several different rating scales to evaluate the severity of dysphagia and other target outcomes (their primary and secondary outcome), the choice of mean difference might not be suitable. To be specific, the mean difference was only suitable for situation that all the included studies applied the same rating scales (or the same units). In contrary, in situation of different rating scales, the standardized mean difference would be a good choice, which could be done by R.
Response: Thanks a lot for your professional comment. Instead of combining all trials in one forest plot, we combined some trials which contained the same dysphagia scales. Thus, mean difference is suitable.
7. I noticed that all the included RCTs applied different polarity of tDCS with different currents. The different polarity and different currents of tDCS might exert different effect (i.e. enhancing or inhibiting effect) to the targeted brain. Please address this issue and how the authors resolve this heterogeneity.
Response: Thank you very much for your insightful comment. First, the I2 statistic was used to evaluate the heterogeneity of the studies. Second, we set many subgroups to detect heterogeneity. Third, sensitivity analyses were also applied to dissect the heterogeneity.
I did not see the figure legend of Figures 3, 4, 5, 6, and 7. Please add it.
Response: We appreciate the reviewer’s rigor concerning this point. We have added an explanation for the annotation to the figure legend.
9. Where is the evaluation of publication bias (i.e. funnel plot or Egger’s test)?
Response: Thanks a lot for your professional comment. We had added the Funnel plots to assessment of publication bias. However, Egger's test for asymmetry in funnel plot would only be performed if 10 or more studies were included. (Result section, Page: 5-6, Line: 89-99)
- Finally, echoing the comment 7, the different polarity/current/target brain regions would contribute to wide variety of effect. However, recently there is another new-raised hypothesis to address the different polarity/current/target brain regions of tDCS management. That is neural noise hypothesis. Please check the reference PMID: 21685932, and PMID: 33691622 and make a brief discussion about this.
Response: Thank you for your helpful guidance. We had made a brief discussion in the Discussion section.(Discussion section, Page: 7, Line:119-133)
11. Finally, per the PRISMA guideline (either 2009 or 2020 version), there are several important items missing. For example, the PRISMA checklist, statement of original authors contact, keywords used in each database, and reasons for exclusion of each study (mandatory for 2020 version). Please revise accordingly.
Response: Thank you for your helpful guidance. I totally agree. the PRISMA checklist and keywords used in each database seen supplement file; statement of original authors contact seen Page: 9, Line: 223-224; reasons for exclusion of each study seen the flowchart of the screening process (Page:5)

Round 2
Reviewer 3 Report
The authors had addressed all my comments. The current version is good to be accepted.